# The Genetics of Glucose-6-Phosphate-Dehydrogenase (G6PD) and Uridine Diphosphate Glucuronosyl Transferase 1A1 (UGT1A1) Promoter Gene Polymorphism in Relation to Quantitative Biochemical G6PD Activity Measurement and Neonatal Hyperbilirubinemia

**DOI:** 10.3390/children10071172

**Published:** 2023-07-06

**Authors:** Arieh Riskin, Yulia Bravdo, Clair Habib, Irit Maor, Julnar Mousa, Sizett Shahbarat, Elena Shahak, Adel Shalata

**Affiliations:** 1Department of Neonatology, Bnai Zion Medical Center, Ruth & Bruce Rappaport Faculty of Medicine, Technion, Israel Institute of Technology, Haifa 32000, Israel; 2Department of Pediatrics, Bnai Zion Medical Center, Ruth & Bruce Rappaport Faculty of Medicine, Technion, Israel Institute of Technology, Haifa 32000, Israel; 3Biochemistry Laboratory, Bnai Zion Medical Center, Ruth & Bruce Rappaport Faculty of Medicine, Technion, Israel Institute of Technology, Haifa 32000, Israel; 4The Simon Winter Institute for Human Genetics, Bnai Zion Medical Center, Ruth & Bruce Rappaport Faculty of Medicine, Technion, Israel Institute of Technology, Haifa 32000, Israeladel.shalata@b-zion.org.il (A.S.)

**Keywords:** glucose-6-phosphate dehydrogenase (G6PD), uridine diphosphate glucuronosyl transferase 1A1 (UGT1A1), neonatal hyperbilirubinemia (NHB), genotype, phenotype, G6PD enzyme activity, G6PD deficiency, *Mediterranean* mutation, *UGT1A1* promoter polymorphism, number of TA repeats

## Abstract

Glucose-6-phosphate dehydrogenase (G6PD) deficiency and polymorphism in uridine diphosphate glucuronosyl transferase 1A1 (UGT1A1) were associated with significant neonatal hyperbilirubinemia (NHB) and increased risk for kernicterus. However, quantitative screening tests for G6PD enzyme activity proved unsatisfactory in estimating the risk for significant NHB, especially in heterozygous females that could present phenotype overlap between normal homozygotes, heterozygotes, and deficient homozygotes, resulting in a continuum of intermediate G6PD activity. Objective: To examine the association of genotype and phenotype in newborns with decreased G6PD activity and its relation to NHB. Study design: Quantitative G6PD enzyme activities were measured on umbilical cord blood samples. After accepting parental consent, samples were analyzed for *G6PD* mutations and *UGT1A1* gene polymorphisms (number of TA repeats in the UGT1A1 promoter). The associations to quantitative G6PD activity and bilirubin levels were assessed. Results: 28 females and 27 males were studied. The *Mediterranean* mutation (*NM_001360016.2(G6PD): c.563C>T (p.Ser188Phe)*) was responsible for most cases of G6PD deficiency (20 hemizygous males, 3 homozygous and 16 heterozygous females). The association between this mutation, decreased G6PD activity and higher bilirubin levels was confirmed. Heterozygosity to 6/7 TA repeats in the *UGT1A1* promoter was associated with increased NHB, especially in female newborns with G6PD deficiency. However, it seems that the interaction between G6PD deficiency, *UGT1A1* promoter polymorphism, and NHB is more complex, possibly involving other genetic interactions, not yet described. Despite genotyping females with G6PD deficiency, the overlap between the upper range of borderline and the lower range of normal G6PD activity could not be resolved. Conclusions: The results of this study highlight the possibility for future implementation of molecular genetic screening to identify infants at risk for significant NHB, especially *UGT1A1* polymorphism in heterozygous females with borderline G6PD deficiency. However, further studies are needed before such screening could be applicable to daily practice.

## 1. Introduction

Neonatal hyperbilirubinemia (NHB) is a clinical condition frequently encountered in newborn infants. Careful evaluation and many times also management [1,2,3] are required, thus making NHB the most common reason for hospital readmission during the first week of life [4]. Although NHB is considered a benign transient physiological phenomenon in many neonates, in few infants total serum bilirubin (TSB) may rise to hazardous levels that pose a direct threat of an acute bilirubin encephalopathy that can lead to kernicterus and brain damage [5,6,7,8,9,10,11,12]. Genetic and environmental factors contribute to the development of NHB [1,2]. However, the important contribution of genetically determined conditions has been increasingly recognized in recent years [13,14,15,16,17,18,19,20,21,22,23,24]. Polymorphism across three genes was particularly reported in association with an increased risk for NHB including those of: (1) the red blood cell enzyme glucose-6-phosphate dehydrogenase (*G6PD*) [13,17,19,20,21,25,26,27,28,29,30]; (2) the hepatic bilirubin-conjugating enzyme uridine-diphosphate glucuronosyl transferase 1A1 (*UGT1A1*) [13,15,16,17,19,20,24,28,30,31,32,33,34,35,36,37,38,39,40,41]; and (3) the hepatic solute carrier organic anion transporter polypeptide 1B1 (*OATP1B1*)—the bilirubin transporter localized to the sinusoidal membrane of hepatocytes, which is the blood-hepatocyte interface that limits bilirubin hepatic uptake [18,19,20,42]. These genetic variants may interact with each other or with environmental contributors to produce significant NHB [19,20]. *UGT1A1* gene was investigated because of its significant role in bilirubin metabolism, namely hepatic bilirubin glucuronidation. The frequent polymorphism described in this gene was the insertion of a seventh (TA) repeat in the promoter sequence of *UGT1A1*, which usually consists of (TA)_6_ repeats. The seventh (TA) repeat in the repetitive (TA)_n_TAA element lessens the affinity of the TATAA binding protein, which is a transcription factor, to the TATAA box18, so that when the number of (TA)_n_ repeats increase above the wild type (TA)_6_, *UGT1A1* expression declines. The (TA)_7_/(TA)_7_ promoter homozygous variant has third of the wild-type UGT1A1 activity, and is responsible for Gilbert syndrome phenotype. Decreased hepatic bilirubin-conjugating capacity due to (TA)_n_ promoter polymorphism was also associated with early-accelerated NHB and prolonged indirect hyperbilirubinemia, particularly in breastfed newborns. It can also increase the risk for significant NHB when coupled with hemolytic conditions. Co-expression of *UGT1A1* variants with other genes was frequently described. As many as two thirds of the individuals who were homozygous for the (TA)_7_ *UGT1A1* promoter variant were also homozygous or heterozygous for the *OATP1B1* polymorphism. The co-expression *OATP1B1* polymorphism with *UGT1A1* variants could result in diminished hepatic bilirubin uptake with decreased hepatic bilirubin conjugation, impairing bilirubin clearance thus increasing hyperbilirubinemia. Many of the G6PD deficient individuals are homozygous or heterozygous to co-expression of the (TA)_7_ variant on at least one allele. Co-expression of bilirubin conjugation-limiting gene variants seem to be important in modulating the risk for NHB, especially when coupled with other risk factors such as G6PD deficiency [19,20,29].

*G6PD* mutations are important contributors to the risk for significant NHB than can even lead to kernicterus [27]. *G6PD* gene variants may predispose to NHB by causing an acute hemolytic event with or without identifiable environmental trigger. Alternatively, *G6PD* mutations can lead to severe NHB by causing low-grade hemolysis coupled with *UGT1A1* gene polymorphisms [17,22,27,28,29,30,35]. *UGT1A1* promoter and coding sequence gene variants may cause significant NHB via decreased hepatic bilirubin conjugation [15,16,17,18,30,31,32,33,34,37,38,39,40,41].

Because G6PD deficiency is an X-linked condition, males may be either G6PD normal or deficient hemizygotes, whereas females may be either normal homozygotes, deficient homozygotes, or heterozygotes. Using biochemical testing, identification of the two male groups should be straightforward. Categorization of females, however, may be inaccurate. In any female cell, only one of the two X chromosomes is active. If the inactivation of the X chromosome was random, half of the cells of a heterozygote female would have been G6PD normal and half would have been deficient, and quantitative G6PD enzyme activity, representing both cell components, would have been intermediate between normal and deficient levels. However, X chromosome inactivation is usually nonrandom, resulting in varying proportions of red blood cells that may be either G6PD normal or deficient [43,44,45]. Thus, in heterozygous females, quantitative measurements of G6PD activities result in a continuum of borderline intermediate levels [46].

Immigration and inter-communal marriages spread G6PD deficiency beyond its original geographic and ethnic distribution. This should be taken into account when assessing risk factors for developing significant NHB. Israel is an immigrant country that absorbed Jews, including Sephardic Jews, coming from all over the world. Native Arabs and Jews originating from many countries comprise most of our population. Thus, our hospital has been one of the first in Israel to implement universal quantitative neonatal screening for G6PD deficiency in order to identify newborns at risk for developing severe NHB [47]. The ethnic characteristics of G6PD deficiency in our newborn population emphasized this approach [47]. A growing population of mixed ethnic origin (Ashkenazi and Sephardic Jewish intermarriages) was found, and 3.8% of the males in this group were G6PD deficient [47]. The World Health Organization (WHO) recommends screening all newborns in populations with a prevalence of 3 to 5% or more in males. Based on the data found on G6PD prevalence in our neonatal population, i.e., 4.5% of all males and even higher in some ethnic groups (10.7% in Sephardic Jews and 6.2% in Muslim Arabs) [47] our center, as well as other birthing medical centers in Israel, adopted a universal screening program for G6PD deficiency. Further justification for universal neonatal G6PD screening was the association between G6PD deficiency and significant NHB, including the increased risk associated with borderline intermediate G6PD activities in female infants [47]. However, the main pitfall of our current universal neonatal screening for G6PD deficiency is that it is a biochemical phenotype-based method. Quantitative measure of G6PD activity still lacks the genotypic equivalent that is so important for defining the infants, especially heterozygous female with intermediate G6PD activities, who may be at high risk for severe NHB and even kernicterus [27], not less than the deficient male infants. Using our universal screening data on quantitative measurements of G6PD activity in our entire population, the sex-based distribution of G6PD ranges of normal, deficient and intermediate activities was assessed [47]. However, the difficult challenge was to define the intermediate borderline range in female newborns. For clinical and practical considerations, adopting a reference value of G6PD activity <7 U/gHb for classifying male newborns as G6PD deficient was useful, although most *Mediterranean G6PD* deficient males in our population were with G6PD activities <2 U/gHb. However, it was hard to define the upper limit of G6PD activity for the intermediate continuum of borderline levels in the presumably heterozygous female groups. The suggested G6PD activity of 9.5 or 10 U/gHb for the upper limit [46,47] was the best approximation that could be achieved based on quantitative G6PD activity measurements without concomitant DNA analysis that was not readily available at that time. 

Thus, the problem of identifying G6PD-deficient newborns at risk for significant NHB was not fully settled. Although most G6PD-deficient males can be accurately identified by quantitatively testing G6PD enzyme activity, females are more difficult to categorize because many in this group may be heterozygotes with phenotype overlap between normal homozygotes, heterozygotes, and deficient homozygotes [44]. Screening females by phenotypic biochemical quantitative enzymatic activity measurement is relatively inaccurate, and requires a wide range of safety zones in order not to miss any of these female infants at risk for severe NHB.

DNA-based polymerase chain reaction (PCR) molecular screening could probably be more accurate, because it identifies the exact genotype of these females. However this is usually more complicated and expensive technology, especially for setting up a screening program [44]. Another difficulty with DNA-based screening is the wide range of worldwide *G6PD* mutations [32,48,49]. Even if only considering the more frequent *G6PD* mutations in our region, dominated by *G6PD^Mediterranean^* that is common in Sephardic Jews and Arabs in our country, a significant number of mutations would still have to be screened for, because of the diverse population, being an immigration country [25,50,51,52,53].

The aim of this study was to examine another strategy to overcome the problem of defining the G6PD borderline deficiency range in relation to NHB. If our phenotypic biochemical quantitative G6PD enzymatic activity screening could be more accurate and reliable in identifying high-risk newborn infants, especially heterozygous females, then the need for genetic screening will decrease. Studying the specific possible genotypes associated with the different levels of G6PD activity in our population and their relations to the development and severity of NHB was thus the approach adopted. The goal of this study was to try to establish G6PD phenotype–genotype associations and relate them to the risk of developing severe NHB. This could be important in order to make our universal screening more clinically informative and practically efficient [44]. Specifically, the aim was to identify *G6PD* gene mutations in our population, identify *UGT1A1* promoter gene polymorphisms, and try to find the association with the biochemical G6PD activity and clinical NHB.

## 2. Materials and Methods

### 2.1. Study Population

Infants studied were born at the Bnai Zion medical center in Haifa. Based on the results of quantitative G6PD screening performed on all newborns in our hospital, male infants with G6PD deficiency (<7 U/gHb) and female infants with low and borderline G6PD levels (<12 U/gHb) were identified. The upper level of 12 U/gHb is beyond the upper limit of intermediate range (10 U/gHb) employed in clinical practice, because one of the study goals was to try and better define the upper limit of intermediate G6PD activity, and 12 U/gHb seemed to be a wide enough range to test. A small sample of male and female infants with normal G6PD activities (>7 U/gHb for males and >12 U/gHb for females) were included as controls.

### 2.2. Study Period

The period of the study was 1 August 2018–30 July 2021. Because of the COVID-19 pandemic with frequent lockdowns, and limited opportunities to obtain informed consents from both parents, sample collection was practically discontinued earlier than planned on 1 March 2020.

### 2.3. Consent

The parents of infants that qualified for genetic testing were approached by one of the first three authors (A.R., Y.B., and C.H.) for informed consent to use the same umbilical sample that was used for G6PD screening for further genetic analysis. All parents that were approached consented for the specific genetic analyses to be performed, as outlined below. However, this cannot be regarded as a cohort of all the infants with G6PD deficiency or borderline deficiency during the study period. Not all parents of infants that qualified for the study were approached because of different issues (i.e., weekends and holidays—if none of the authorized researchers was available to come and discuss consent with the parents; or pauses in collection of samples due to workload or technical issues in the involved laboratories).

### 2.4. Ethics

The study was approved by the local institutional review board (Helsinki committee) (0117-17-BNZ) on 23 July 2018 after also being approved by the supreme national review board of the Israeli Ministry of Health (Application number 044-2018) on 29 March 2018.

### 2.5. Measured Parameters

G6PD phenotype, i.e., G6PD activity, *G6PD* genotype, i.e., *G6PD* mutation (*Mediterranean* or other) and homozygosity or heterozygosity, *UGT1A1* gene polymorphisms, i.e., the number of TA repeats in the *UGT1A1* promoter, and bilirubin levels (either by non-invasive transcutaneous bilirubinometry or in serum from blood sample) were measured.

### 2.6. Study Design

Our suggested approach for studying G6PD genotype–phenotype relations and their association to NHB involved the following stages:Developing a genetic methodology (high-resolution melting (HRM) analysis) that would enable us to easily identify mutant *G6PD* males and heterozygous and homozygous females compared to normal wild type *G6PD* controls.Establishing a method to identify *UGT1A1* gene polymorphisms (number of TA repeats in the *UGT1A1* promoter) in normal and G6PD-deficient infants, and establish their associations with NHB in G6PD-deficient infants.After establishing the method for identifying *G6PD* mutations in our population and their associations to specific *UGT1A1* gene polymorphisms associated with NHB, we checked the *G6PD* genotype in a sample of newborn infants, including females, who were allegedly heterozygotes. An infant’s G6PD status was first identified by our universal umbilical cord blood’s G6PD enzyme activity screening. For the purpose of this study, for female infants with intermediate G6PD activity measurements, the upper limit of intermediate range was widened beyond what is currently used (2–12 U/gHb). Before genetic testing, the parents of these infants were asked for their informed consent to use their infants’ umbilical cord sample (taken in EDTA tube useful both for G6PD enzyme activity testing and for DNA mutation analysis) for genotype testing as part of this study.Bilirubin levels of the infants studied were closely followed in order to identify whether they develop NHB and to define its severity. In this stage, our aim was to try to establish the phenotype–genotype relationships between the intermediate range of biochemically measured G6PD enzyme activity, the specific *G6PD* mutation, and the *UGT1A1* promoter gene polymorphisms that could identify infants, especially females, at risk for significant NHB.

### 2.7. Sample Size

For stages 1 and 2, recruitment of 50 infants was planned including infants with G6PD deficiency or borderline deficiency in a male: female ratio of 1:1, and ~15% infants with normal G6PD activity (>7 U/gHb for males (4) and >12 U/gHb for females (4)). For stage 3, recruitment of another 50 infants was planned: 10% males with G6PD deficiency (<7 U/gHb (5)) and 90% females with intermediate (2–12 U/gHb, higher upper level as discussed above) deficiency (45). As mentioned above, recruitment was discontinued on 1 March 2020 at the beginning of stage 3, before the desired sample size was reached. However, failure to establish HRM in our population as a fast low-cost method for identifying G6PD heterozygous and homozygous mutants without having to identify the exact mutation by sequencing also made efforts at continuation of recruitment to stage 3 after the end of the pandemic irrelevant.

### 2.8. Procedures

**Collection of blood samples for universal G6PD screening:** Universal screening for G6PD was implemented at the Bnai Zion Medical Center since July 2007 [47]. After delivery, umbilical cord blood samples for quantitative G6PD activity screening are routinely obtained. These are collected in ethylene diamine tetra acetic acid (EDTA) tubes that can also be used for DNA extraction and PCR analysis.
**Biochemical laboratory assays:**
2.1G6PD enzymatic activity was measured within two days of collection. Red cell G6PD activity was determined by the enzymatic colorimetric assay for quantitative determination of G6PD deficiency using a commercial kit (G6P-DH, Cat. No. 17.005, Sentinel Diagnostics, Milan, Italy). The quantitative test involves oxidation of glucose-6-phosphate to 6-phosphogluconate with concomitant reduction of NADP+ to NADPH. The rate of NADPH formation, which is proportional to G6PD activity, is measured spectrophotometrically. The G6P-DH screening kit also contains Hemoglobin Normalization procedure, i.e., a rapid quantitative measurement of G6PD activity is coupled to a simultaneous evaluation of hemoglobin content. Results are expressed as units of activity per gram hemoglobin (U/gHb). All the tests of enzymatic activities were automatically run at 36 °C by the biochemistry analyzer, Cobas Mira (Roche diagnostic systems, Hoffman La-Roche LTD, Basel, Switzerland).2.2Total serum bilirubin (TSB) levels were spectrophotometrically determined by Twin Beam Analyzer (Gamidor Diagnostics Ltd., Petach Tikva, Israel,) at two wavelengths (455 nm and 575 nm).
**Transcutaneous bilirubinometry and clinical assessment of NHB:** Transcutaneous bilirubin (TcB) measurements were performed using the Minolta JM-105 (Dräger Jaundice meter—Biliblitz). TcB was measured routinely in all newborns at the time of discharge from the nursery (usually 52 ± 12 h after uncomplicated normal vaginal delivery). TSB was also measured if the TcB reading was higher than 11 mg/dL or the baby had known risk factors for significant neonatal hyperbilirubinemia. If both TcB and TSB were recorded, TSB was used for the analysis. Assessment of the severity of hyperbilirubinemia and decisions regarding phototherapy were based on the AAP guidelines [1] that were adopted by the Israeli Neonatal Association.**Genetic laboratory analysis:** After obtaining parents’ informed consent, DNA was extracted from the collected blood samples saved in EDTA tubes.4.1Complete PCR sequencing of the whole *G6PD* gene was performed, including the coding and the one non-coding exons, flanking intronic regions of the G6PD gene, and the *G6PD* gene 5′ and 3′ untranslated regions (5′UTR, 3′UTR). A list of the G6PD primers that were used can be found in the Appendix A. In this method, one primer is used with dNTP and ddNTP nucleotides. ddNTP nucleotides are used as transmitters of the sequencing reaction. Because discontinuation of elongation is random, multiple fragments of DNA with different lengths are synthesized—each strand was stopped at a different point. A mixture of four ddNTP nucleotides, each carrying fluorophore with different fluorescent color, is used. At the stage of electrophoresis in the sequencer, the fluorescent signal is measured during the passage through the capillary in the optical cell. The information is electronically recorded, and the translated sequence is saved to the computer electronic database. Because this complete PCR sequencing is both time and cost consuming, initially it was planned only for a small number of representative samples—two of our control group were sequenced in order to verify a normal sequence. After that, the rest of the samples along with the control group were checked by GeneScan.4.2For all samples, employment of a new methodological approach to identify *G6PD* heterozygous and homozygous mutants using high resolution melting (HRM) analysis was planned. The advantage of using this method could be identification of *G6PD* heterozygous and homozygous mutants without identifying the exact mutation, as performed in sequencing, which should be faster and much less time and cost consuming. HRM is a new post-PCR analysis method used to identify genetic variation in nucleic acid sequences. It can discriminate DNA sequences based on their composition, length, GC content, or strand complementarity. HRM analysis starts with PCR amplification of the region of interest in the presence of a dsDNA-binding dye, which has high fluorescence when bound to dsDNA. The second step is a high-resolution melting step and capturing fluorescent data points per change in temperature. When the dsDNA dissociates into single strands, the dye is released which causes a change in fluorescence. Finally, a melt curve profile of the amplicon is received. Melt curves with similar shape but different melting temperature (Tm) represent homozygous variant samples compared to a wild type sample. Melt curves with different shape are due to heterozygous variant samples. Tm is the point in the melt curve where 50% of the DNA is double-stranded and 50% is single-stranded (melted); it is visualized better in a first derivative curve (as a peak). After aligning the samples, the result is plotted and presented as the pre-melt region (100% fluorescence where every amplicon is double-stranded), active melt region (true fluorescence change), and post-melt region (0% fluorescence point where every amplicon is single-stranded). The differences between melt curves are often small and are best visualized using a difference plot that helps distinguish between the homozygous and heterozygous compared to the wild type. In order to detect unknown mutations, the whole gene was scanned. Using HRM analysis, scanning of the gene using 180 bp long amplicons with 25-mer forward and reverse primers and 25 bp overlap of each amplicon was initially done.4.3Initial HRM analysis of the first 10 male samples, although technically not optimal, compared with the melting curve of the normal *G6PD* sequence control group, revealed that the melting curve of exon 5 was different. Taking this into account and supported by published literature [54], exon 5 was directly sequenced and hemizygosity of the known *Mediterranean* mutation (*NM_001360016.2(G6PD): c.563C>T (p. Ser188Phe)*) was found in 6 out of 10 samples. Therefore, Sanger direct sequencing of exon 5 was the first step in all tested samples thereafter.4.4In order to find an association between the number of TA repeats (microsatellites) in the *UGT1A1* promoter and quantitative G6PD enzyme activity, especially in the heterozygous females, the number of TA repeats in the *UGT1A1* promoter was determined. Wild type *UGT1A1* contains six TA repeats [A(TA)_6_TAA] in its promoter region [31]. Seven or more TA repeats was considered pathological. The analysis was performed using a sequencing method with primers that were specifically designed and synthesized for this purpose.


### 2.9. Statistical Analysis

Data were statistically analyzed using SigmaPlot, version 11.0 (Systat Software Inc. San Jose, CA, USA). Statistical analysis included descriptive statistics, and one-way analysis of variance (ANOVA) or chi-square test for comparisons of multiple continuous or categorical variables between groups. When appropriate, the non-parametric test of Kruskal–Wallis one-way analysis of variance on ranks was employed. Data are presented as mean ± standard deviation (SD) or median with interquartile range (IQR), as appropriate, and *p*-values of less than 0.05 are considered statistically significant.

## 3. Results

Fifty-five newborn infants were included in the study: 28 females and 27 males.

The *G6PD* gene has three different isoforms, as demonstrated in Figure 1, including 13 exons, and they differ in the first exon sequence, which is encoding only in *NM_000402.4(G6PD)* isoform.

According to the professional and public databases, there are 238 known damaging *G6PD* variants in the Human Gene Mutation Database (HGMD) and 299 pathogenic and 23 likely pathogenic variants in ClinVar (https://www.ncbi.nlm.nih.gov/clinvar, accessed on 12 June 2023). In our research group, the *NM_001360016.2(G6PD): c.563C>T (p.Ser188Phe)*, known as the *Mediterranean* pathogenic variant, was by far the most frequent mutation identified in our population, both in males and females. The *Mediterranean* pathogenic variant is also known as *NM_000402.4(G6PD):c.653C>T (p.Ser218Phe)* as explained in Figure 1 and illustrated at the genomic level in Figure 2 (30 codons shift (p.Ser188Phe p.Ser218Phe) due to exon 1 encoding only in isoform *NM_000402.4*).

In the 27 male newborn infants’ group, 20 males were hemizygous to the *Mediterranean* mutation (*NM_001360016.2(G6PD): c.563C>T (p. Ser188Phe)*) and seven did not have this mutation. All 20 hemizygous males with the *Mediterranean* mutation had very low G6PD activity of less than 1 U/gHb. The number of TA repeats in the *UGT1A1* promoter were 7/7 in three (abnormal homozygous), nine were heterozygous with 6/7 repeats, and eight were normal with 6/6 repeats. Among the seven males who were not found to have the *Mediterranean G6PD* deficiency mutation, three had G6PD activity lower than 7 U/gHb (in the range of 5–6.5). In this group of three, full sequencing of the *G6PD* gene was performed and other mutations of G6PD were found. Two of these infants were normal with 6/6 TA repeats in the *UGT1A1* promoter, and one was heterozygous with 6/7 repeats. The four other male infants without the *Mediterranean* or any other *G6PD* mutation had normal G6PD activity above 7 U/gHb. Three of these infants were normal with 6/6 TA repeats and one was heterozygous with 6/7 TA repeats in the *UGT1A1* promoter (Figure 3).

Of the 28 female newborn infants studied, three were homozygous to the *Mediterranean* mutation with low G6PD activity of 0–1.4 U/gHb in the biochemical quantitative assay. Genetically, two of these females were heterozygous with 6/7 TA repeats in the *UGT1A1* promoter, and the third was normal with 6/6 TA repeats (Figure 3).

Sixteen females were heterozygous to the *Mediterranean* mutation, 15 of them had no other *G6PD* mutation. Their G6PD activity was in the range of 6.2–11.6 U/gHb. Seven of this group had normal TA repeats (6/6) in the *UGT1A1* promoter, six were heterozygous with 6/7 repeats, and for the other two the blood sample was insufficient to complete this analysis and their data are missing. One female of the 16 *Mediterranean* heterozygous infants was also heterozygous to another *G6PD* mutation, defining her as compound heterozygous. Her G6PD activity was low—2 U/g Hb, and her genetic analysis revealed 6/7 TA repeats in the *UGT1A1* promoter (Figure 3).

Nine of the 28 female infants did not have the *Mediterranean* mutation. Eight of them had no other *G6PD* deficiency mutation. Their G6PD activity in the biochemical assay was 8.8–18.2 U/gHb. Genetic analysis of the *UGT1A1* promoter revealed that three were homozygous with abnormal 7/7 TA repeats, two were heterozygous with 6/7 TA repeats and three were normal with 6/6 repeats. The last female in this group was homozygous for another *G6PD* mutation. Her G6PD activity was 6.3 U/gHb and UGT1A1 promoter analysis revealed 6/6 normal TA repeats (Figure 3).

Regarding G6PD activity, hemizygous male infants with the *Mediterranean* mutation had significantly lower G6PD activity. Although hemizygous males with another *G6PD* mutation demonstrated lower G6PD activity compared to males with normal G6PD activity, the difference was not statistically significant, most probably due to the relatively small group of such males in our sample (Table 1).

Homozygous and heterozygous G6PD-deficient females had lower G6PD activity compared to females without *G6PD* mutations. The homozygous females had low G6PD activities, and the heterozygous females had intermediate activities. The differences were statistically significant. Most of the *G6PD*-deficiency mutations were *Mediterranean*, as presented above. The few female infants with other *G6PD* mutations were subdivided for the analysis. The one who was compound heterozygous, i.e., heterozygous to two *G6PD* mutations, one of which was *Mediterranean*, was addressed as homozygous for G6PD deficiency. Another female who was homozygous to another (non-Mediterranean) *G6PD* mutation was analyzed with the female infants who were heterozygous to the *Mediterranean* mutation having intermediate G6PD activity (Table 1).

There was some overlap between the upper range of intermediate G6PD activity (6.2–11.6 U/gHb) in heterozygous female infants and lower range of normal G6PD activity (9.7–18.2 U/gHb) in female infants without mutations.

Although there were no significant differences in mean maximal bilirubin measured during nursery admission between the different genotypic *G6PD* groups of male and female newborns, there were some noteworthy differences (Table 1). Mean bilirubin level measured in the homozygous female infants was 9.3 mg/dL with levels that could reach as high as 20 mg/dL. Mean bilirubin level in the heterozygous females was slightly lower at 8.7 mg/dL with levels up to 18.5 mg/dL. In the group of females without *G6PD Mediterranean* mutation, mean bilirubin level was lower at 7.4 mg/dL with a highest bilirubin level of 11.0 mg/dL measured in one of the infants (Table 1). Mean maximal bilirubin levels in the male infants were also not significantly different. In the group of 20 males who were hemizygous for the *Mediterranean* mutation, the highest bilirubin measured was 17.0 mg/dL as opposed to 12.2 mg/dL in the males with the normal *G6PD* genotype (Table 1).

Analyzing the number of TA repetitions in the UGT1A1 promoter revealed that the highest bilirubin levels were found in the two males hemizygous to the *Mediterranean* mutation who were also homozygous to 7/7 TA repeats (mean: 12.1 ± 0.3 mg/dL); and in the male who was hemizygous to another non-Mediterranean *G6PD* mutation who was heterozygous to 6/7 TA repeats (14.8 mg/dL). However, these were very small groups and the differences were not statistically significant (Table 2).

In the females, heterozygosity to 6/7 TA repeats in the *UGT1A1* promoter was associated with the highest bilirubin levels (mean 10.1 ± 5.6 mg/dL, compared to 7.6 ± 4.1 in females with wild type *UGT1A1*, i.e., homozygous to 6/6 TA repeats in the promoter, and 5.4 ± 1.9 in homozygous with 7/7 TA repeats, *p* = 0.438) (Table 2).

Heterozygosity to 6/7 TA repeats in the UGT1A1 promoter in females was also associated with lower G6PD activity (mean 6.9 ± 4.0 U/gHb vs. 15.3 ± 4.1 in homozygous to 7/7 TA repeats and 9.5 ± 4.3 in homozygous to 6/6 TA repeats, *p* = 0.032).

## 4. Discussion

The *Mediterranean* pathogenic variant (*NM_001360016.2(G6PD): c.563C>T (p.Ser188Phe)*), also known as *NM_000402.4(G6PD):c.653C>T (p.Ser218Phe)* as explained in the results above (Figure 1 and Figure 2), was associated in both hemizygous males and homozygous females with low G6PD activity. Heterozygous females with the *Mediterranean* mutation exhibited intermediate range G6PD activity with higher maximal bilirubin levels that were slightly lower than in the homozygous females with the mutation. Thus, the known association between the *Mediterranean G6PD* mutation and lower G6PD activity with higher levels of neonatal hyperbilirubinemia was confirmed [51].

Heterozygosity to 6/7 TA repeats in the promoter of *UGT1A1* was associated with more significant neonatal jaundice, especially if the newborn infant had complete or partial G6PD deficiency. This association seems to be more significant in females. However, from our analyses it seems that the interaction between G6PD deficiency and decreased conjugation by UGT1A1 because of incorrect number of TA repeats in the promoter with resultant increased neonatal hyperbilirubinemia without hemolysis [35] is more complex, and is possibly associated with other genetic interactions that have not yet been described [24].

However, it must be stressed that in this study only *UGT1A1* variants that had different polymorphisms in the promoter, and more specifically in the number of TA repeats in the (TA)_n_TAA box of the promoter, were investigated [32,35,41]. The many other variants of the *UGT1A1* gene, which could significantly affect its activity and thus NHB, were not addressed [16,24,33]. Although Israel is an immigrant country with multiple diverse populations, many ethnic populations are under-represented or not represented in our population. There are many variants of *UGT1A1* and *G6PD* that have been described in the world [28,31,36,37,38,39,40] and were not addressed in this study; thus, their co-expression and effect on NHB deserve further studies in order to clarify the complex interactions that cause significant NHB.

Neonatal screening programs for G6PD increase parental and caretaker awareness, thereby facilitating early access to treatment with resultant diminished mortality and morbidity associated with severe NHB. Regarding our aim to try and better define the intermediate borderline G6PD activity range in females that are heterozygous to G6PD deficiency, so that these female infants could be better identified and their parents be guided before discharge from the nursery regarding the need for close follow-up for late-onset rapidly rising hyperbilirubinemia on days 4–6 of life, the genetic analysis was not helpful. Our results still demonstrate the overlap between the higher range of intermediate G6PD activity and the lower range of normal. Actually, in the range of 9.7–11.6 U/gHb there remains uncertainty that mandates caution in females with G6PD activity measured within this range. Thus, the threshold of normal G6PD activity in females, defined as 9.5 by Kaplan et al. [46,55] or 10 U/gHb by Riskin et al. [47], based on the distribution of G6PD activities measured by the biochemical method and the relation to NHB remains in doubt.

The main limitation of our study is our sample size that was affected by many factors including the costs of full genetic analysis. This was the result of our failure to establish HRM as a quick cheap method to identify *G6PD* mutations in our population instead of running full sequencing of the *G6PD* gene. This was quite surprising because HRM was successfully used in the past to identify *G6PD* mutations [56], including in our region [57]. Yet, it is possible that the combination of dominant *Mediterranean G6PD* mutation in our population along with the other *G6PD* mutations typical of our population as an immigrant country resulted in melting curves that were not sensitive enough to separate the mutations by HRM.

Regarding the possible contribution of analyzing the number of TA repeats in the *UGT1A1* promoter as another screening method to identify newborns, especially females, at risk for high NHB, it seems that the interaction with G6PD deficiency and significant neonatal jaundice is more complex, and requires further study and analysis before it can be recommended as a screening test for newborns. Recent studies highlight the independent role of the number of TA repeats in the *UGT1A1* promoter in NHB [24,38].

## 5. Conclusions

In summary, it is important to continue performing more studies to evaluate the role of genetic screening with analysis of the number of TA repeats in the *UGT1A1* promoter to assess for increased risk for NHB, especially in heterozygous females to G6PD deficiency. However, it cannot be recommended routinely at this time before more data and better interpretation of the genetic interactions are achieved.

## Figures and Tables

**Figure 1 children-10-01172-f001:**
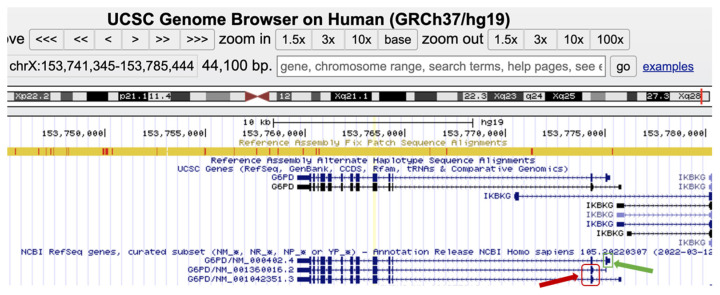
Screenshot of the *G6PD* different isoforms, *NM_000402.4, NM_001360016.2*, and *NM_1042351.3*, as illustrated in https://genome.ucsc.edu/, accessed on 12 June 2023. The *G6PD* isoforms differ in exon 1 where, exon 1 of NM_000402.4 is encoding (green arrow), while, in the other two isoforms exon 1 is not encoding. The first ATG start codon of both isoforms is located in the second exon (red rectangle). This fact can explain the difference in G6PD enzyme amino acid residue nomenclature in the *NM_000402.4* isoform compared to the *NM_001360016.2* and *NM_1042351.3* isoforms.

**Figure 2 children-10-01172-f002:**
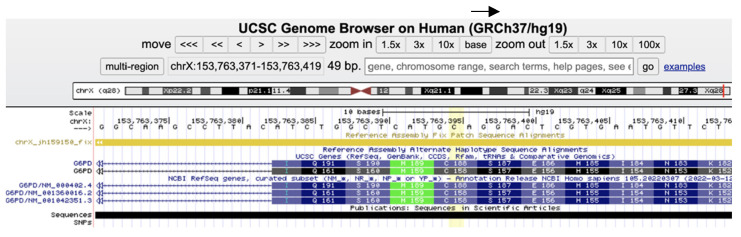
Screenshot of the Mediterranean *G6PD* mutation as demonstrated in https://genome.ucsc.edu/, accessed on 12 June 2023); the *NM_001360016.2(G6PD): c.563C>T (p. Ser188Phe)* (yellowish column) with its specific isoform nomenclature.

**Figure 3 children-10-01172-f003:**
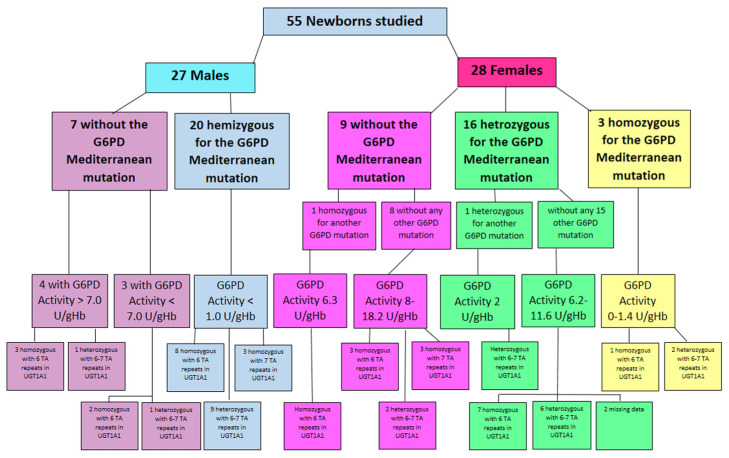
Study population by gender, *G6PD* mutation, G6PD activity, and number of TA repetitions in the *UGT1A1* promoter.

**Table 1 children-10-01172-t001:** G6PD activity in male and female newborns by genetic profile and its association to maximal bilirubin levels measured in these infants.

Gender	n	*G6PD* Genotype	G6PD Activity (U/gHb)		*p*-Value	Maximal Bilirubin Level			*p*-Value
			Median	IQR		Median	IQR	Maximal Value	
**Male**	20	Hemizygous to the *Mediterranean G6PD* mutation	0.60 *	0.25–0.80	<0.001 *	9.35	7.45–11.85	17.0	0.138 ^¶¶^
	3	Hemizygous for another *G6PD* mutation	5.20	5.05–5.87		12.90	10.95–14.32	14.8	
	4	Normal	16.55 *	15.50–17.30		9.65	9.35–10.95	12.2	
**Female**	4	Homozygous **	1.00 ^†^	0.30–1.70	<0.001 ^†¶^	8.55	3.75–14.85	20.1	0.839 ^¶¶^
	16	Heterozygous ^††^	8.40 ^¶^	7.35–10.65		8.70	4.40–12.65	18.5	
	8	Normal	13.30 ^†¶^	10.15–17.10		6.60	5.00–10.35	11.0	

IQR—Interquartile range (25–75% percentile). *—Kruskal–Wallis one-way analysis of variance on ranks (significantly different groups are marked by *). ^†^—Kruskal–Wallis one-way analysis of variance on ranks (significantly different groups are marked by ^†^). **—This group includes one female who was heterozygous for the Mediterranean mutation and also heterozygous for another G6PD mutation, i.e., compound heterozygous. ^††^—This group includes one female who was homozygous for another (non-Mediterranean) mutation. ^¶^—Kruskal–Wallis one-way analysis of variance on ranks (significantly different groups are marked by ^¶^). ^¶¶^—Kruskal–Wallis one-way analysis of variance on ranks.

**Table 2 children-10-01172-t002:** G6PD genotype and UGT1A1 polymorphism in male and female newborns and their association to maximal bilirubin levels measured in these infants.

Gender	N	G6PD Genotype	UGT1A1 Promoter	n	Maximal Bilirubin Level		
			Number of TA Repeats		Median	IQR	*p*-Value
**Male**	20	Hemizygous to the Mediterranean G6PD mutation	6/6	9	9.50	7.82–12.30	0.199 *
			6/7	9	8.60	6.45–10.25	
			7/7	2	12.15	11.90–12.40	
	3	Hemizygous for another G6PD mutation	6/6	2	11.60	10.30–12.90	
			6/7	1	14.80	–	
	4	Normal	6/6	1	9.10	–	
			6/7	3	9.70	9.62–11.57	
**Female**	4	Homozygous **	6/6	1	7.50	–	0.698 *
			6/7	3	9.60	2.40–17.48	
	16	Heterozygous ^†^	6/6	8	6.45	4.40–9.80	
			6/7	6	11.75	8.10–13.10	
			Unknown	2	8.60	3.70–13.50	
	8	Normal	6/6	3	5.60	5.52–8.82	
			6/7	2	10.90	10.80–11.00	
			7/7	3	4.50	4.20–6.82	

IQR—Interquartile range (25–75% percentile). *—Kruskal–Wallis one-way analysis of variance on ranks. **—This group includes one female who was heterozygous for the Mediterranean mutation and also heterozygous for another G6PD mutation, i.e., compound heterozygous. ^†^—This group includes one female who was homozygous for another (non-Mediterranean) mutation.

## Data Availability

Data is unavailable due to privacy and ethical restrictions as dictated by the Helsinki committees.

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
