# Peer review of "The Genetics of Glucose-6-Phosphate-Dehydrogenase (G6PD) and Uridine Diphosphate Glucuronosyl Transferase 1A1 (UGT1A1) Promoter Gene Polymorphism in Relation to Quantitative Biochemical G6PD Activity Measurement and Neonatal Hyperbilirubinemia"

_children, 2023, doi:10.3390/children10071172_

Round 1
Reviewer 1 Report
Congratulation to the authors for the great work done. Detailed comments about this study are as follows:
-In Table 1, the authors compared maximal bilirubin levels across three groups of the G6PD genotype using a one-way ANOVA on means. However, there are a large standard deviation of the homozygous G6PD genotype in female (mean +/- SD = 9.3 +/- 8.3) and the heterozygous G6PD genotype in female (mean +/- SD = 8.7 +/- 4.8), they are considered that non-normal distribution. Therefore, the Kruskal-Wallis test might be more suitable than ANOVA.
Author Response
We want to thank the reviewer for the thoughtful and helpful comments.
Following are our responses to the reviewer's comments:
-In Table 1, the authors compared maximal bilirubin levels across three groups of the G6PD genotype using a one-way ANOVA on means. However, there are a large standard deviation of the homozygous G6PD genotype in female (mean +/- SD = 9.3 +/- 8.3) and the heterozygous G6PD genotype in female (mean +/- SD = 8.7 +/- 4.8), they are considered that non-normal distribution. Therefore, the Kruskal-Wallis test might be more suitable than ANOVA.
We have corrected our statistical analysis as the reviewer suggested. The corrected analysis appears in Table 1.

Reviewer 2 Report
This article describes the association of genotype and phenotype in new-borns with decreased glucose-6-phosphate dehydrogenase and its relation with high levels of bilirubin.
Overall, the article is confusing to read as authors describe a rather simple process. There is a lot of unnecessary repetition that makes the paper longer and more difficult to read. The readability should be improved in general, as several sections are too verbose.
I have several observations:
1. Genes must be written in italics.
2. Line 58, line 65 and line 67: Please discuss or talk more about these references. Some perspective is missing on what is already in previous studies and how each polymorphism influences the patient.
3. Line 75: Add reference.
4. In scientific articles it is strongly recommended to be written in an impersonal way. Avoid the use of "we".
5. The introduction is confusing and focuses on article 44 by the same authors, the purpose or justification of the project is not detailed in depth.
6. Material and methods: Please follow an orderly structure: 1. Study population: patients, study period, consent, and ethics committee. 2. Measured parameters: state the parameters that are going to be measured and how they are going to be measured. 3. Genetic analysis: explain how the analysis of G6PD and UGT1A1 is carried out.
7. Line 172-198: It is not a protocol, scientific articles must summarize the information, it looks like a copy paste of the request to the ethics committee more than a scientific article.
8. There is a lot of repeated information about the sample, the size, and that makes it difficult to read and maintain a common thread.
9. 2.6.4.1: Please explain which primers are used and from which laboratory. Have they been designed by the authors?; are they reference primers? There is a lack of references on the methodology when something is established.
10. Results: Very disorganized results. Line 313-319 should be put in methodology.
11. Figure 1: Is illegible, lower font size.
12. Line 397-402: it would be interesting to put this results in a table.
13. Discussion: Figure 2 and 3 are unnecessary and are part of the results.
14. The discussion is very brief, the results are not compared with other articles. What does your study bring new? Is there agreement with previous results in other populations with these polymorphisms?
15. Lastly, more than 50 of the references are older than 5 years. I want to suggest that the authors update these references where possible.
Author Response
We want to thank the reviewer for the thoughtful and helpful comments.
Following are our responses to the reviewer's comments:
Genes must be written in italics
This was corrected.
Line 58, line 65 and line 67: Please discuss or talk more about these references. Some perspective is missing on what is already in previous studies and how each polymorphism influences the patient.
We added a short discussion on the different polymorphisms, their co-expression and their effect on hyperbilirubinemia (lines 62-83).
Line 75: Add reference.
References were added (now line 100).
In scientific articles it is strongly recommended to be written in an impersonal way. Avoid the use of "we".
We thoroughly reviewed the paper and changed the text to impersonal way, as suggested.
The introduction is confusing and focuses on article 44 by the same authors, the purpose or justification of the project is not detailed in depth.
Indeed, our previous article on this subject (now 47) and the debate it raised in the medical literature was a strong drive for doing this research. We felt that we have clearly outlined the rationale for doing this study and its goals. However, based on the reviewer's comment we reviewed the introduction, and especially its last paragraph, in order to clarify our study aims.
Material and methods: Please follow an orderly structure: 1. Study population: patients, study period, consent, and ethics committee. 2. Measured parameters: state the parameters that are going to be measured and how they are going to be measured. 3. Genetic analysis: explain how the analysis of G6PD and UGT1A1 is carried out.
We have revised the methods section based on the reviewer's comments. However we felt that the section on the 'Study design' was important for further understanding of this study aims, and how we planned to deal with these study questions.
Line 172-198: It is not a protocol, scientific articles must summarize the information, it looks like a copy paste of the request to the ethics committee more than a scientific article.
Please refer to our previous answer. Although a scientific article, this study and specifically in this journal is aimed also for clinicians, and for this audience we felt that the study design or protocol clarifies our methodological approach to the study questions even for the clinicians who are not involved in laboratory studies regularly.
There is a lot of repeated information about the sample, the size, and that makes it difficult to read and maintain a common thread.
The issue of our sample size and calculations of it for both study stages are important, especially in light of the fact that our main study limitation was our sample size, and the fact that we did not get as many samples as we initially planned. Thus, it is fully explained. The presentation of the methodology in numbered sections allows the readers who are not interested in this information to skip this paragraph without having problem following the rest of the manuscript.
2.6.4.1: Please explain which primers are used and from which laboratory. Have they been designed by the authors?; are they reference primers? There is a lack of references on the methodology when something is established.
A list of the G6PD primers we used was added as a supplemental material for this manuscript submission (a note referring the readers to this list is written in lines 282-283).
Results: Very disorganized results. Line 313-319 should be put in methodology.
Lines 313-319 were moved to the methodology section (now lines 327-334 in the Methods section).
Figure 1: Is illegible, lower font size.
This was the largest font size that could fit into this figure (now Figure 3). We must admit that here we took advantage of the journal being on line. Readily available PDF zoom of 150-175% makes this figure easily readable. Also, all the text in the figure is also addressed in the text of the results if not fully clear from the figure itself.
Line 397-402: it would be interesting to put this results in a table.
We willingly accepted this excellent suggestion by the reviewer, and added Table 2 with all the data.
Discussion: Figure 2 and 3 are unnecessary and are part of the results.
Figures 2-3 were necessary for the less experienced reader who might be confused by the different nomenclature of the Mediterranean mutation. However, we agree with the reviewer that this discussion should be part of the results, and was thus moved to the beginning of the Results section (now Figures 1-2).
The discussion is very brief, the results are not compared with other articles. What does your study bring new? Is there agreement with previous results in other populations with these polymorphisms?
We have discussed the issues raised by the reviewer. However, based on the suggestions of the reviewer we added more references to our comparisons, and added a full paragraph regarding the complex interactions of co-expression of the different gene variants, and their possible effects on hyperbilirubinemia. We also addressed the issue of comparison to other populations that was limited because we studied only our limited population, diverse as it may be.
Lastly, more than 50 of the references are older than 5 years. I want to suggest that the authors update these references where possible.
We did a thorough literature research before and during this study and the preparation of this manuscript, and along with old references, there are also new ones. Many of the breakthroughs in the research of G6PD and UGT1A1 polymorphism were done in the late 90's and especially in the first decade of the 21st century, and we think they need to be quoted when re-examining these issues. It is also a tribute of honor for the leading researchers who made this remarkable progress in our understanding of these issues. Indeed there is still many unanswered or not fully resolved questions, but the basic concepts are supported on the grounds of these important studies. Our understanding is that when we present these complex issues to the audience of the journal, especially for the clinicians, we need to support our manuscript with a relatively detailed introduction that is fully referenced (sort of a mini-review). I hope the editors and the reviewer will accept our approach regarding this.

Round 2
Reviewer 2 Report
The authors took into consideration all the suggestions proposed and I think that now the paper can be published however some minor revisions are still needed:
Check that all genes are in italics, some are correct, other are not: tables, footer...